# ATP-Binding Cassette G Transporters and Their Multiple Roles Especially for Male Fertility in *Arabidopsis*, Rice and Maize

**DOI:** 10.3390/ijms23169304

**Published:** 2022-08-18

**Authors:** Suowei Wu, Chaowei Fang, Ziwen Li, Yanbo Wang, Shuangshuang Pan, Yuru Wu, Xueli An, Yan Long, Xiangyuan Wan

**Affiliations:** Zhongzhi International Institute of Agricultural Biosciences, Shunde Graduate School, Research Center of Biology and Agriculture, University of Science and Technology Beijing (USTB), Beijing 100024, China

**Keywords:** ABCG transporter, anther and pollen development, vegetive and female organ development, stress response, phytohormone transport, substrate analysis, *Arabidopsis*, rice, maize

## Abstract

ATP-binding cassette subfamily G (ABCG) transporters are extensive in plants and play essential roles in various processes influencing plant fitness, but the research progress varies greatly among *Arabidopsis*, rice and maize. In this review, we present a consolidated nomenclature and characterization of the whole 51 ABCG transporters in maize, perform a phylogenetic analysis and classification of the ABCG subfamily members in maize, and summarize the latest research advances in ABCG transporters for these three plant species. ABCG transporters are involved in diverse processes in *Arabidopsis* and rice, such as anther and pollen development, vegetative and female organ development, abiotic and biotic stress response, and phytohormone transport, which provide useful clues for the functional investigation of ABCG transporters in maize. Finally, we discuss the current challenges and future perspectives for the identification and mechanism analysis of substrates for plant ABCG transporters. This review provides a basic framework for functional research and the potential application of ABCG transporters in multiple plants, including maize.

## 1. Introduction

ATP-binding cassette (ABC) transporter, one of the largest proteins family, widely exists in bacteria, animals, and plants [1]. ABC transporters are involved in countless cellular processes, including the transmembrane transport of various unrelated molecules [2]. ABC proteins possess nucleotide-binding domain (NBD) which contain several highly conserved motifs, including the Walker A and B sequences, the ABC signature motif, the H loop and the Q loop; and transmembrane domains (TMDs), each composed of several hydrophobic α-helices [1,3]. ABC proteins can be full-size (with two TMDs and two NBDs) or half-size (with one TMD and one NBD), in which case they may function as homodimer or heterodimer [2]. The two NBDs cooperate to bind and hydrolyze ATP, providing the energy for transport, and the TMDs are involved in substrate recognition and translocation across the lipid bilayer [3]. In plants, ABC proteins can be divided into eight subfamilies, including ABCA to ABCG, and ABCI. Among them, the ABCG subfamily is the largest one in plants, and there are 43 and 54 members in *Arabidopsis* and rice, respectively [3]. The outstanding diversification of ABCGs is thought to be associated with plant adaptation to land environment [4]. In *Arabidopsis* and rice, ABCG transporters have been widely studied and linked with anther and pollen development, cuticle formation, defense response, hormone transport, and seed germination [2]. The substrates of ABCG transporters, including lipids, hormones, and heavy metal ions, etc., have been unambiguously identified using isotope labeling experiments in vivo, or indirectly by imprecise measurements using the GC-MS system [5]. However, the actual substrates of most ABCG transporters remain uncharacterized.

It has been reported that 54 ABCG members exist in maize [6], but only three ABCG transporters (ZmMS2/ZmABCG26, ZmMS13 and ZmGL13) have been characterized and reported to be required for male fertility or leaf cuticle formation in maize [7,8,9,10]. Notably, the current nomenclature of the maize ABCG subfamily does not comply with the standard naming system used in *Arabidopsis* and rice [3,6], and thus results in a lot of inconveniences and problems in maize ABCG research.

Here, we present a consolidated nomenclature system for the maize ABCG subfamily, and focus on functional research progress of the reported ABCG genes in *Arabidopsis* and rice, and their implications for maize orthologs. The functions of maize ABCG genes can be predicted by homologous and bioinformatic analyses. Furthermore, we summarize the current progress and challenges in the identification of ABCG substrates, and present the future research directions and potential applications for ABCG proteins in plants.

## 2. The ABCG Transporters in Maize and Their Phylogenetic Analysis

### 2.1. The Nomenclature and Characterization of the Maize ABCG Subfamily

A consolidated nomenclature for maize ABCG transporters will provide much needed clarity and a framework for future research. Based on BLAST searches against the maize B73 reference genome (B73 RGV4.0) in the MaizeGDB database (https://maizegdb.org/ (accessed on 8 May 2022)), a total of 51 ABCG genes were identified in maize, and named as ZmABCG1 to ZmABCG51 according to their domain organizations (including 31 half-size ABCGs with one NBD-TMD and 20 full-size ABCGs with two NBD-TMD) and chromosome locations in maize (Table 1). Among them, 34 ZmABCGs have synonyms in MaizeGDB, including different members with the same name, such as two members (Zm00001d028689 and Zm00001d013960) named ABCG2, four members (Zm00001d032601, Zm00001d002871, Zm00001d048621 and Zm00001d020811) named ABCG11, etc., which will lead to confusion in the functional research of maize ABCG transporters in the future. Thus, it is desirable to create a unified nomenclature before the confusing and repeated ABCG names proliferate further.

Notably, the total number (51) of maize ABCG genes is different from that (54) in the previous report [6], which may have resulted from an alternative splicing of three ABCG genes, including *ZmABCG5*, *ZmABCG7* and *ZmABCG12* (Appendix A). Furthermore, the spatial expression patterns of maize ABCG genes, based on RNA-seq data analyses in MaizeGDB, are shown in Table 1. These will provide useful information for investigating their biological functions. Other characteristics of the 51 ZmABCG genes, including the corresponding gene models in B73 reference genome (B73v3), genome physical locus, topology, and subcellular localization prediction using Cell-PLoc 2.0 (http://www.csbio.sjtu.edu.cn/bioinf/Cell-PLoc-2/ (accessed on 20 April 2022)), are listed in Appendix A. All of the 51 ZmABCG genes are distributed across the 10 chromosomes of maize, varying in numbers from eight on chromosome 8, to two on chromosome 5 (Table 1, Figure 1). The chromosome location information of ABCGs will be helpful in exploring their function and evolution in the future.

### 2.2. Phylogenetic Analysis and Classification of ABCGs

In order to analyze the evolutionary relationship of ABCGs in maize, rice and *Arabidopsis*, the maximum likelihood method and the MEGA7 program were used to construct the phylogenetic tree, based on the multiple alignment of all the ABCG protein sequences in these tree plant species. All the identified ABCGs were divided into two subgroups in the phylogenetic tree (Figure 2). Subgroup 1 contains 90 half-size ABCGs, including 31 in maize, 31 in rice and 28 in *Arabidopsis*, which can be further classified into five clades (Clades 1-1 to 1-5). Among them, some ABCGs have been reported to be involved in anther and pollen development, such as AtABCG9 [11], AtABCG26 [12,13,14], OsABCG15 [15,16] and ZmABCG27 (ZmMS2) [8,10] in Clade 1-1, AtABCG11 [17], OsABCG26 [18,19], ZmABCG13 (ZmMS13) [7] in Clade 1-2, AtABCG1/16 [20,21,22], and OsABCG3 [23,24] in Clade 1-4. Notably, the ABCGs in Clade 1-4 can be classified into two subclades of monocots and dicots, indicating that diversification of these ABCG members is later than that between monocots and dicots during plant evolution. Subgroup 2 consists of 58 full-size ABCGs, including 20 in maize, 23 in rice and 15 in *Arabidopsis*, which can be further classified into three clades (Clades 2-1 to 2-3). Considering that some of the ABCGs have been functionally characterized in *Arabidopsis* and rice and several maize orthologs of *Arabidopsis* and rice, known ABCGs have been reported to play similar roles during anther development and male fertility, such as the orthologous ZmABCG27 (previously named as ZmMS2/ZmABCG26) [8,10], OsABCG15 [15,16] and AtABCG26 [12,13,14], the orthologous ZmABCG13 (previously named as ZmMS13) [7], OsABCG26 [18,19] and AtABCG11 [17]. Therefore, the phylogenetic analysis result will provide helpful information in exploring the functions of unknown ABCG genes in maize.

## 3. Multiple Functions of ABCG Transporters in *Arabidopsis*, Rice and Maize

To date, at least 30 ABCG genes in *Arabidopsis* and 11 ABCG genes in rice have been functionally characterized, but only three ABCG genes have been identified in maize. These reported ABCG transporters play essential roles in various biological processes, such as anther and pollen development, vegetive and female organ development, biotic and abiotic stress response, and phytohormone transport and signaling (Table 2; Figure 3).

### 3.1. Anther and Pollen Development

ABCG transporters are involved in the translocation of various cuticular wax and cutin monomers, sporopollenin precursors, and tryphine components, from the tapetum where they are generated, to the anther outer surface and locule for anther cuticle and pollen wall formation, respectively (Figure 3A). Loss-function mutations of some ABCG transporters often lead to defective anther cuticle and pollen wall formation and thus lead to male sterility in plants.

To date, at least 12 ABCG genes have been reported to be required for anther/pollen development and male fertility in multiple plant species (Table 2). For example, *Arabidopsis* AtABCG11 acts as both homodimer or heterodimer with other ABCGs (e.g., AtABCG5, 9, 12 and 14), and plays multiple roles in flower cuticle formation, root suberin metabolism, and proper vascular development. The *abcg11* mutant displays dwarfism, male sterility, post-genital organ fusions, and reduced cutin load in flowers [17,25,26]. OsABCG26 and ZmMS13, orthologs of AtABCG11 in rice and maize, are essential for anther cuticle and pollen exine formation, especially for the translocation of the cuticular wax and cutin monomers generated in the tapetum across anther wall layers for anther cuticle formation [7,18,19]. AtABCG26 and its orthologs OsABCG15 in rice and ZmMS2 in maize, play similar roles in controlling pollen exine development and male fertility by mainly transferring sporopollenin precursors from the tapetum onto the developing microspore surface, although loss-of-function mutations of OsABCG15 and ZmMs2 also lead to defective anther cuticle formation [8,10,15,16]. *Arabidopsis* AtABCG1 and AtABCG16, and their rice ortholog OsABCG3, are required for the transport of nexine and intine precursors for male gametophyte development in the post-meiotic stages, and for pollen tube growth; the *atabcg1atabcg16* double mutant and *osabcg3* mutant show defective nexine and intine formation, and thus male fertility [21,22,23,24]. Furthermore, AtABCG9 and AtABCG31 are involved in the specific transport of steryl glycosides from the tapetum for pollen coat (tryphine) deposition [11]. Most recently, Liu et al. has reported that AP1/2β adaptins mediated exocytosis of tapetum-specific ABCG transporters (such as AtABCG9 and AtABCG16) from the trans-Golgi network (TGN) to the plasma membrane, and are required for pollen development in *Arabidopsis* [70]. AtABCG28 is specifically expressed in mature pollen grains and pollen tubes, and is critical for localizing polyamines [precursors of reactive oxygen species (ROS)] at the growing pollen tube tip in *Arabidopsis* [31]. Together, ABCG transporters may play conserved and divergent roles in transferring various precursors essential for anther development and male fertility in monocots (rice and maize) and dicots (*Arabidopsis*).

### 3.2. Vegetative and Female Organ Development

Additionally, several ABCG transporters are reported to be involved in the formation of diffusion barriers, such as cuticle, suberin, and lignin, during the vegetative and female organ development in plants (Table 2, Figure 3B). For example, AtABCG5, a half-size transporter of cutin and wax precursors, is required for the dense cuticle layer formation in young seedlings, forming homodimer or heterodimers with AtABCG11, and possibly with AtABCG12 and other ABCG transporters [33]. AtABCG11, but not AtABCG12, forms a homodimer to transport wax and cutin precursors, and AtABCG11 forms a heterodimer with AtABCG12 to transport wax precursors from the cuticle layer in *Arabidopsis* [26]. AtABCG13 is required for the secretion of flower cuticular lipids, particularly in petals and carpels [37], similar to the full-size AtABCG32 transporter, which exports particular cutin precursors from the epidermal cell in leaves and flowers [43]. AtABCG27 and AtABCG33 may be involved in cellulose synthesis, based on expression analysis using *Arabidopsis* cell suspensions during tracheary element differentiation [40]. Furthermore, the orthologs of *Arabidopsis* AtABCG32, rice OsABCG31 [45] and maize ZmGL13 [9] are also essential for leaf cutin and cuticular wax formation. Notably, four half-size *Arabidopsis* ABCG transporters, AtABCG1, AtABCG2, AtABCG6, and AtABCG20 [21,27], as well as their rice ortholog OsABCG5/RCN1 [34] are responsible for the transport of suberin monomers in roots and seed coats, and OsABCG5/RCN1 is also essential for shoot branching, by promoting the outgrowth of lateral shoots [35]. AtABCG29 acts as a monolignol (p-coumaryl alcohol) transporter involved in lignin biosynthesis [42]. Collectively, ABCG transporters play indispensable roles in the formation of various diffusion barriers in plants, which are critical for plant growth and the development against various stresses.

### 3.3. Biotic and Abiotic Stress Response

Another important function of ABCG transporters is in the protection against biotic and abiotic stresses (Table 2, Figure 3C). Many ABCG transporters are involved in pathogen and other biotic stress responses by the secretion of defense molecules. For example, AtABCG34 mediated the secretion of camalexin to defend against the necrotrophic pathogens *Alternaria brassicicola* and *Botrytis cinerea* [47], and AtABCG36/PEN3 was reported to enhance resistance against several non-adapted pathogens [58]. AtABCG36/PEN3 and AtABCG40/PDR12 mediate camalexin secretion for the resistance against *Botrytis cinerea* [63]. The half-size ABC transporters STR1 and STR2 are indispensable for mycorrhizal arbuscule formation in rice [64]. Increasing reports reveal that ABCG transporters perform important physiological functions in plant drought stress response by regulating stomatal closure, such as OsABCG5/RCN1 [50], AtABCG17 and AtABCG18 [67], AtABCG21 and AtABCG22 [39,68], AtABCG25 [69], and AtABCG40 [62], while OsABCG9 plays a critical role in the transportation of epicuticular wax and is essential for rice drought response [53]. Overexpression of the AtABCG19 transporter confers kanamycin resistance to transgenic plants, which may be related to zinc homeostasis in plants [51,52]. Furthermore, AtABCG35, AtABCG36 and their rice ortholog OsABCG36, as well as OsABCG43, are involved in cadmium tolerance and other heavy metal stresses [54,55,56,60]. Together, ABCG transporters play critical roles in various biotic and abiotic stress responses in plants by transporting different substrates.

### 3.4. Hormone Transport and Signaling

In addition, ABCG transporters are also involved in phytohormone transport, which is very important for plant growth and development, including seed germination, shoot development, root formation, stress response, and other physiological processes [71]. To date, at least four types of hormones transported by 14 transporters have been identified in *Arabidopsis* and rice (Table 2, Figure 3D). Among them, *Arabidopsis* AtABCG14 is essential for the long-distance translocation of cytokinins from root to shoot [65], and its ortholog, OsABCG18, plays a similar role in rice and promotes grain yield [66]. Several ABCGs have been identified as ABA transporters that are required for the long-distance translocation of ABA in *Arabidopsis* and rice, such as AtABCG17 and AtABCG18 [67], AtABCG22 [68], AtABCG25 [69], AtABCG30, AtABCG31, and AtABCG40 [30,62], and OsABCG5 [50]. While auxin and its precursors (IBA) are reported to be transported by ABCG transporters, including AtABCG1 and AtABCG16 [20], AtABCG36 [59], and AtABCG37 [49], the ABCG transporters AtABCG1 and AtABCG16 are also involved in subcellular distribution in the metabolism and signaling of Jasmonates [29].

In summary, some ABCG transporters play different roles during plant growth and development, as they may function as homodimers and/or heterodimers with different ABCGs and thus transport various substrates.

## 4. Functional Predictions of ABCG Genes in Maize

Compared with *Arabidopsis* and rice, less ABCG genes have been functionally identified in maize (Table 2). Considering the functional conservation of orthologs during plant evolution, maize ABCG genes might also play similar roles with their orthologs, as with *Arabidopsis* and rice. Moreover, the spatiotemporal expression patterns of genes are often associated with their biological functions. Therefore, homologous and bioinformatics analyses will provide useful information for exploring the function of maize ABCG genes.

### 4.1. Functional Prediction of ABCGs Based on Homologous Analysis

Multiple studies have shown that some ABCG orthologs of *Arabidopsis*, rice, and maize play both conserved and divergent roles in regulating anther development and male fertility. For example, ZmABCG27/ZmMS2 and its orthologs OsABCG15 and AtABCG26, are all required for the translocation of sporopollenin lipidic precursors from the tapetum to the locules for pollen exine development in maize, rice, and *Arabidopsis* [8,10,13,14,15,19]. ZmABCG13/ZmMS13 is essential for anther cuticle and pollen exine formation, consistent with its orthologs AtABCG11 and OsABCG26 in *Arabidopsis* and rice [7,17,19]. Here, we list the maize orthologs of all identified ABCG genes in *Arabidopsis* and rice (Table 2), which will provide an important clue for the functional characterization of maize ABCG genes by reverse genetic strategies, such as CRISPR/Cas9 or RNAi. For example, given that AtABCG9 and AtABCG31 are essential for the deposition of steryl glycosides on the pollen coat, and thus for pollen fitness [11], and *OsABCG3* and its orthologs *AtABCG1*/*16* are required for pollen wall (nexine and intine layers) formation [18,21,22,24], their orthologs in maize might also be involved in pollen wall development, which needs to be confirmed in the future via reverse genetic strategies. Therefore, it is feasible to predict the functions of more unknown ABCG genes in maize using homologous analysis.

### 4.2. Functional Prediction of ABCGs Based on Bioinformatic Analysis

The expression analysis based on RNA sequencing (RNA-seq) data provides a new window to predict gene function. Based on anther RNA-seq data, there are 62 putative maize genic male-sterility (GMS) genes, 125 putative lipid metabolic GMS genes, and 112 putative sugar metabolic GMS genes which have been predicted in maize [72,73,74]. Additionally, some of these predicted GMS genes have been verified through CRISPR/Cas9 mutagenesis in maize [8,75]. Thus, bioinformatic analysis, such as RNA-seq data analysis, provides an effective way to predict the functions of unknown genes.

Here, we carried out transcriptomic analyses of all maize ABCG members based on the RNA-seq data from four maize inbred lines W23, B73, Oh43, and Zheng58 developing anthers (Figure 4A). Based on the RNA-seq data analyses, the 51 maize ABCG genes can be classified into two clusters (I and II). Cluster I can be further divided into four subclusters: I-1 to I-4. Subcluster I-1 consists of six ABCG genes, and half of them (ZmABCG22, ZmABCG8 and ZmABCG24) show peak expressions at middle anther stages (S8 to S9–10), and half of them (ZmABCG10, ZmABCG11 and ZmABCG34) display peak expressions at late anther stages (S11 or S12), in agreement with the qPCR results (Figure 4B1–B6). Interestingly, the rice ortholog (OsABCG3) of ZmABCG8 and ZmABCG24, and *Arabidopsis* ortholog (AtABCG31) of ZmABCG34 have been reported as being required for pollen wall development and male fertility [11,21], indicating that these ABCGs are most likely involved in anther and pollen development in maize. Subcluster I-2 and I-3 include eight and 26 ABCG members, respectively, with relatively low expression during different anther developmental stages. Notably, ZmABCG26 and ZmABCG14 in subcluster 1–3, show an anther-specific expression pattern based on the RNA-seq data retrieved from the MaizeGDB website (http://www.maizegdb.org/ (accessed on 20 April 2022)), and they are orthologs of AtABCG28 required for pollen tube growth and male fertility in *Arabidopsis* [28], indicating that ZmABCG26 and ZmABCG14 might also be required for pollen tube growth in maize. This needs to be proven by reverse genetics. Subcluster I-4 includes three ABCG genes with multiple expression peaks in developing anthers (Figure 4B7–B9), including ZmABCG2 which is orthologous to AtABCG1/16 [21,22], suggesting its potential roles in pollen wall development. Cluster II covers eight ABCG genes with relatively high expression during anther development [7,8,10] (Figure 4B10–B15), including four orthologs of the known GMS genes in *Arabidopsis* and rice, such as ZmABCG27/ZmMS2 orthologous to AtABCG26 and OsABCG15 [8,10,13,14,15,19], ZmABCG13/ZmMS13, ZmABCG3 and ZmABCG6 orthologous to AtABCG11 and OsABCG26 [7,17,19], and two of them (ZmABCG27 and ZmABCG13), but not ZmABCG3, have been confirmed to be required for male fertility in maize [7,8,10], implying that these orthologous ABCG genes play both conserved and diversified roles during anther development among different plants.

## 5. Substrate Identification of Plant ABCG Transporters

Although ABCG transporters take part in various physiological processes in plants, the exact substrates of most ABCG transporters are still unclear. The functional insights into plant ABCG transporters are mainly gained from reverse genetics (e.g., CRISPR/Cas9 mutagenesis) and metabolite analysis. However, the phenotypic effects resulting from knockout studies might be pleiotropic, thus hindering the identification of the actual substrates.

The most powerful approach to identify the substrates of ABCG transporters is to use a transport assay which demonstrates translocation activity across a membrane in a strictly ATP-dependent manner. The prerequisite for the transport assay is the overexpression of the ABCG protein in a proper expression system, such as tobacco (*N. benthamiana*) protoplast system and yeast strains (YMM12 and BY-2) cell lines [30,44,49]. To date, substrates of ten *Arabidopsis* ABCG transporters have been identified by employing radioactivity labeled compounds, i.e., by using isotope labeling experiments in vivo (Table 3). For example, *Arabidopsis* AtABCG11 and AtABCG32 are reported to export cutin precursors (e.g.,10,16-diOH, C16:0-2-glycerol and W-OH C16:0) for plant cuticle formation, based on the export assay using the protoplast system of *N. benthamiana* [44]. Four AtABCG transporters collaboratively deliver ABA from the endosperm to the embryo for controlling seed germination: AtABCG25 and AtABCG31 export ABA from the endosperm, whereas AtABCG30 and AtABCG40 import ABA into the embryo [30]. However, many isotope labeling compounds are not commercially available and these experiments are time consuming, which hinders their extensive utilization and efficiency.

An alternative approach is the substrate analysis of ABCG mutant and wild-type plant tissues using the GC-MS system. Here, a rational guess about the substrate or substrate class can be made by using the metabolite analysis of specific plant tissues, such as anthers, roots and leaves. The substrates of at least 15 ABCG transporters in *Arabidopsis*, rice and maize have been predicted based on this approach (Table 3). One advantage of this approach is that mixtures, such as the content of the plant’s cytosol, can be directly employed, and subsequently, the isolated compounds can be used as direct proof. Thus, this is a very powerful approach in identifying the substrate of ABCG transporters, although the conclusion is not very convincing. For example, ZmMS13 and ZmMS2 encoding ZmABCG13 and ZmABCG27, respectively, are reported to be essential for the transport of anther cuticle and sporopollenin precursors, partly due to the lipidomic analysis of the wild-type and mutant mature anthers, using GC-MS [7,10].

Besides the two approaches mentioned above, other direct or indirect transport assays have also been reported to have been used to identify the substrates of ABCG transporters. For example, the lipidic substrates of AtABCG1 and ZmMS13/ZmABCG13 were indicated by the ATPase assay of purified protein in vitro [7,27], Cytokinin was identified as the substrate of OsABCG18 based on an export assay with a heterologous expression of OsABCG18 in the protoplast system of *N. benthamiana* [66], and AtABCG28 was required for the apical accumulation of reactive oxygen species in growing pollen tubes based on the immunostaining of polyamines in the growing tip of pollen tubes [31] (Table 3).

Notably, although the substrate identification of plant ABCG transporters has gained some encouraging progress, the translocated substrates and the detailed transport mechanisms of the majority of ABCG proteins remain unclear, which need to be investigated in the future.

## 6. Conclusions and Perspective

ABCG transporters, as one of the largest subfamilies of ABC transporters, play critical roles in various processes influencing plant fitness, especially in plant reproductive and vegetative organ development, hormone transport and stress response. Compared with the unified named ABCG transporters in *Arabidopsis* and rice [3], the ABCG transporters in maize lack a standard naming system, leading to confusion in the functional research of maize ABCGs. In this review, we firstly established a consolidated nomenclature and chromosome location map of the 51 ABCG transporters in the maize genome (Table 1 and Figure 1), providing a basic framework for future research. Based on the phylogenetic analysis, the whole ABCG subfamily members in *Arabidopsis*, rice and maize can be divided into two subgroups and eight clades (Figure 2). These results will provide useful clues for exploring the functions of unknown ABCG genes in maize. Secondly, we summarized the latest research progress of ABCG transporters present in *Arabidopsis*, rice, and maize. The functions of the reported ABCG transporters are involved in diverse processes, such as anther and pollen development, vegetative and female organ development, abiotic and biotic stress response, and phytohormone transport (Table 2 and Figure 3). Thirdly, compared with the plentiful in-depth studies of ABCGs in *Arabidopsis* and rice, less ABCG transporters in maize have been functionally characterized to date. We thus provide two methods for the functional prediction of ABCG transporters in maize, namely the homologous and bioinformatic analyses, based on the functional conservation of ABCG orthologs during plant evolution and spatiotemporal expression patterns of ABCG genes in maize (Table 1 and Figure 4). Finally, we summarize and discuss the current approaches and challenges in the substrate identification of plant ABCG transporters, including the transport assays, by employing radioactivity compounds in vivo, metabolite analysis, using the GC-MS system, and other direct or indirect transport assays (Table 3), although the actual substrates of most plant ABCG transporters remain largely unknown.

In summary, given the tremendous progress in defining the critical roles of plant (mainly in *Arabidopsis* and rice) ABCG transporters involved in various biological processes influencing male fertility and plant fitness, it is plausible that the corresponding ABCG orthologs might also play similar roles in other crops, including maize. This can be verified by the use of reverse genetics (e.g., CRISPR/Cas9 mutagenesis analysis), and thus deepen our understanding of the functional mechanism of ABCG transporters in multiple plants. The derived mutant lines with elite characters, such as male sterility or higher fitness, have potential applications in hybrid crop breeding and seed production in the future.

## Figures and Tables

**Figure 1 ijms-23-09304-f001:**
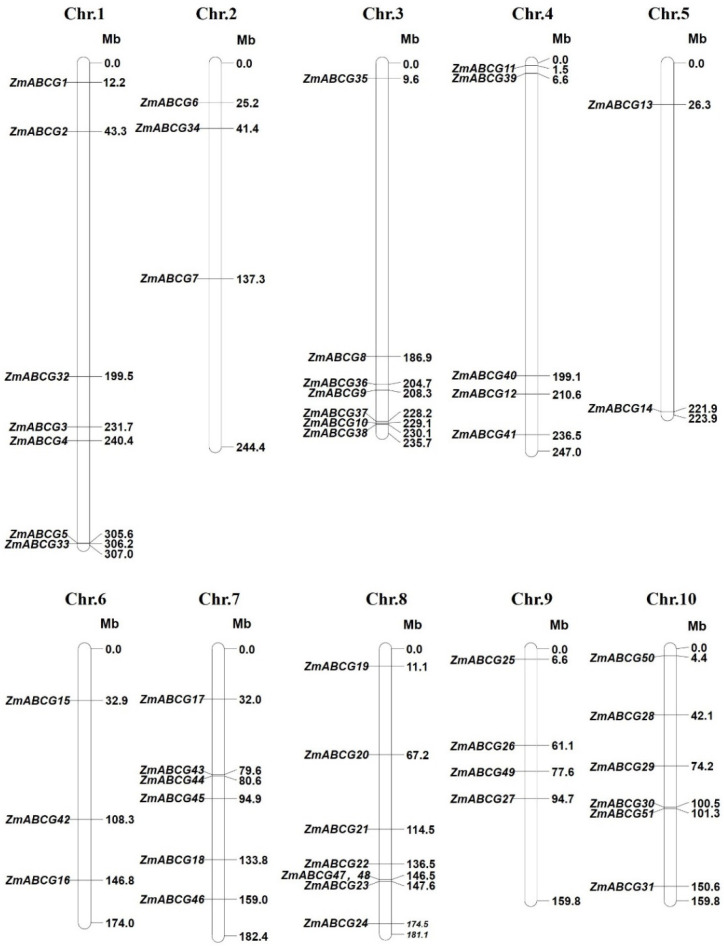
Genomic distribution of the 51 *ZmABCG* genes in maize chromosomes. Chromosome number and genomic length are indicated at the top and bottom of each chromosome, respectively. The physical location of each *ZmABCG* gene is indicated to the right of chromosomes. Mb, million base pair.

**Figure 2 ijms-23-09304-f002:**
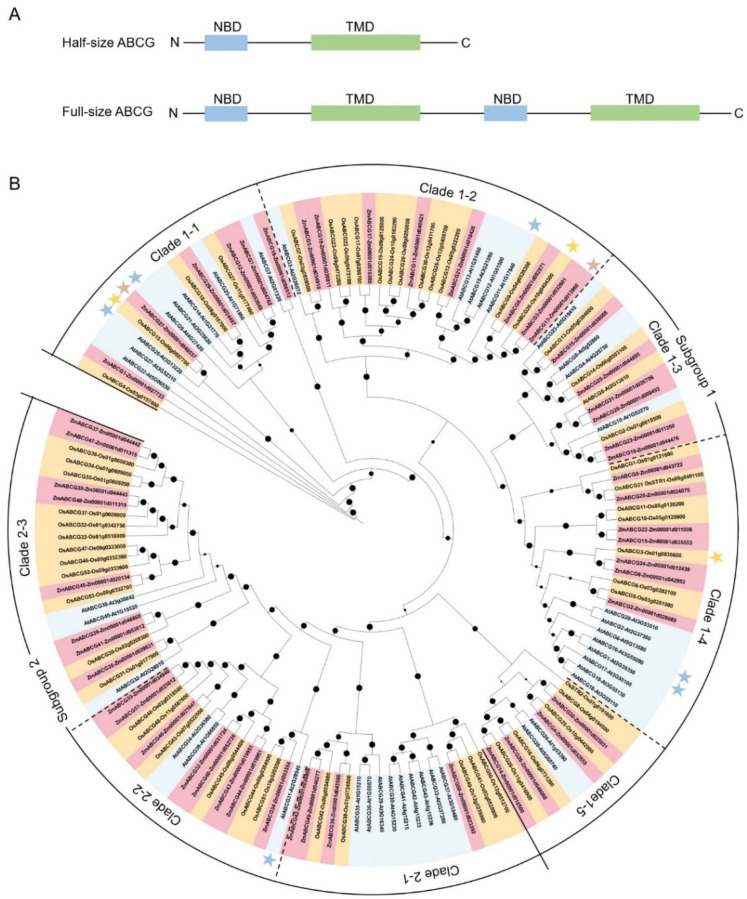
Phylogenetic analysis of ABCGs in maize, rice and *Arabidopsis*. (**A**) Gene structures of half-size and full-size ABCG. NBD, nucleotide binding domain; TMD, transmembrane domain; (**B**) Phylogenetic analysis of ABCGs in maize (51), rice (54) and *Arabidopsis* (43). Subgroup 1 contains 90 half-size ABCGs, including 31 in maize, 31 in rice and 28 in *Arabidopsis*. Subgroup 2 consists of 58 full-size ABCGs, including 20 in maize, 23 in rice and 15 in *Arabidopsis*. Subgroups 1 and 2 can be classified into 5 and 3 clades, respectively. Stars indicate the ABCGs involved in male sterility.

**Figure 3 ijms-23-09304-f003:**
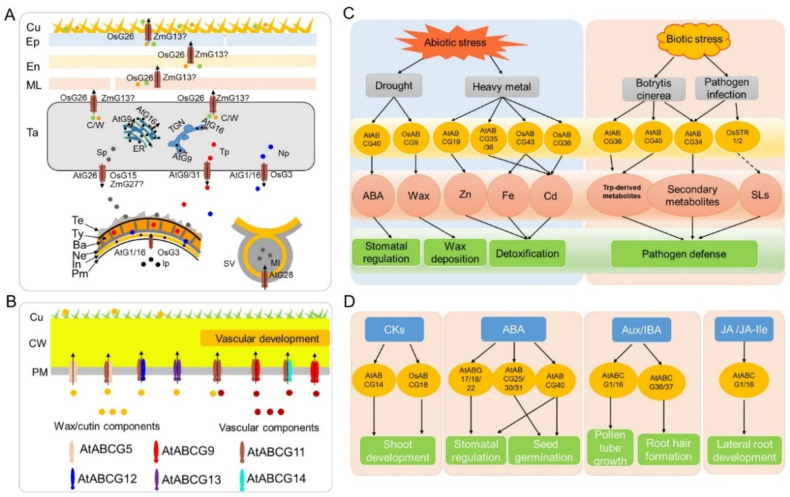
The main functions of ABCG transporters in *Arabidopsis*, rice and maize. (**A**) The proposed working model of ABCG transporters involved in anther and pollen development. Several ABCGs transfer tapetum-generated lipidic and other precursors for anther cuticle and pollen wall development, and AtABCG28 is required for the apical accumulation of reactive oxygen species in growing pollen tubes. AtG, AtABCG; Ba, baculum; Cu, cuticle; C/W, cutin and wax; Ep, epidermis; En, endothecium; In, intine; Ip, intine precursors; MI, membrane lipids; ML, middle layer; Ne, nexine; Np, nexine precursors; Pm, plasma membrane; Sp, sporopollenin precursors; SV, secretory vesicle; Ta, tapetum; Te, tectum; TGN, trans-Golgi network; Ty, tryphine; Tp, tryphine precursors; ZmG, ZmABCG. (**B**) The working model of ABCG transporters contributed to the formation of a tight cuticle layer and vascular development in *Arabidopsis*. The homodimers of AtABCG5/AtABCG5, AtABCG11/AtABCG11, and AtABCG13/AtABCG13, and the heterodimers of AtABCG11/AtABCG5 and AtABCG11/AtABCG12 transfer wax/cutin components for the formation of tight cuticle layers. The homodimers of AtABCG11/AtABCG11 and AtABCG9/AtABCG9, and the heterodimers of AtABCG11/AtABCG9 and AtABCG11/AtABCG14 transport vascular components for vascular development. Cu, cuticle; CW, cell wall; PM, plasma membrane. (**C**) Regulation of plant resistance by ABCG transporters in response to biotic and abiotic factors. The stress factors, ABCG transporters and their corresponding substrates and physiological roles in plants, are displayed in (**C**). ABA, abscisic acid; SLs, strigolactones. (**D**) Hormone transport and signaling of ABCG transporters. The hormone, ABCG transporters and their biological functions are displayed in (**D**). CKs, cytokinins; ABA, abscisic acid; Aux, Auxin; IBA, indole-3-butyric acid; JA, jasmonic acid; JA-Ile, jasmonic acid-isoleucine.

**Figure 4 ijms-23-09304-f004:**
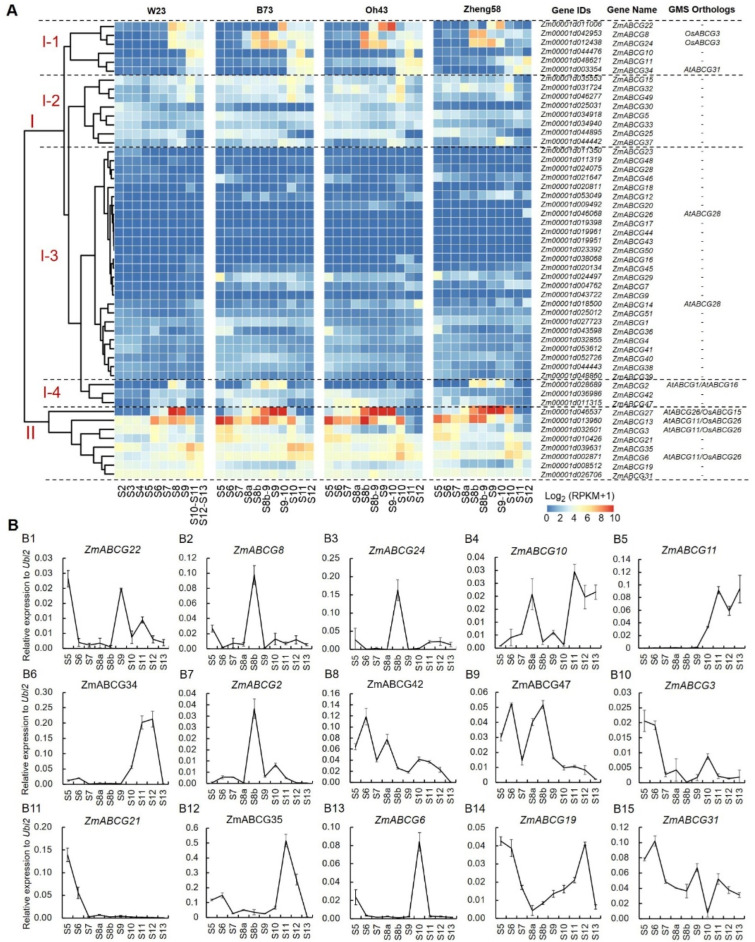
Expression analysis of ABCG genes in maize. (**A**) Expression analysis of 51 ABCG genes in maize based on RNA-seq data in maize inbred lines W23, B73, Oh43 and Zheng58. These genes are clustered into two clusters. Genes in cluster I were clustered into four sub-clusters. (**B**) qPCR analysis of 15 ABCG genes from sub-clusters I-1, I-2, and cluster II in developing anthers from stages 5 to 13 (S5–S13). Data are means ± SD, n = 3.

**Table 1 ijms-23-09304-t001:** The maize ABCG Subfamily: new nomenclature based on B73 reference genome V4.0 and pre-existing synonyms.

No.	New Name	Synonyms ^1^	Gene Models (B73 V4)	Chr.	Length (aa)	Expression Organs ^2^									
						Root	Meristem	Leave	Internode	Tassel	Anther	Cob	Silk	Seed	Embryo
1	ZmABCG1	ABC-2	Zm00001d027723	1	753			+					+		
2	ZmABCG2	ABCG2	Zm00001d028689	1	790			+		+					
3	ZmABCG3	ABCG11	Zm00001d032601	1	736		+	+	+	+			+	+	+
4	ZmABCG4	ABCG24	Zm00001d032855	1	883	+	+	+	+	+	+		+	+	+
5	ZmABCG5	ABCG3	Zm00001d034918	1	1078	+	+	+	+	+	+		+	+	+
6	ZmABCG6	ABCG11	Zm00001d002871	2	721			+	+	+		+	+	+	
7	ZmABCG7	ABCG25	Zm00001d004762	2	632	+	+	+	+						
8	ZmABCG8	ABCG6	Zm00001d042953	3	761	+		+		+					
9	ZmABCG9	ABCG25	Zm00001d043722	3	738			+	+	+					
10	ZmABCG10	ABCG10	Zm00001d044476	3	609	+		+		+					
11	ZmABCG11	ABCG11	Zm00001d048621	4	720			+		+			+		+
12	ZmABCG12	-	Zm00001d053049	4	636	+	+	+	+			+	+	+	
13	ZmABCG13	ABCG2	Zm00001d013960	5	710			+	+	+	+	+	+	+	+
14	ZmABCG14	ABCG24	Zm00001d018500	5	1112						+				
15	ZmABCG15	ABCG16	Zm00001d035553	6	714	+	+	+	+	+	+			+	+
16	ZmABCG16	-	Zm00001d038068	6	694	+		+		+					
17	ZmABCG17	-	Zm00001d019398	7	693	+		+	+	+					
18	ZmABCG18	ABCG11	Zm00001d020811	7	736	+		+		+					
19	ZmABCG19	ABC-2	Zm00001d008512	8	695		+	+	+	+		+	+	+	+
20	ZmABCG20	-	Zm00001d009492	8	275	+	+	+	+			+		+	+
21	ZmABCG21	-	Zm00001d010426	8	690			+	+	+	+	+		+	+
22	ZmABCG22	-	Zm00001d011006	8	712	+	+	+	+	+	+		+	+	+
23	ZmABCG23	ABCG10	Zm00001d011350	8	507	+		+							
24	ZmABCG24	ABCG16	Zm00001d012438	8	743	+		+		+					
25	ZmABCG25	-	Zm00001d044895	9	618	+	+	+	+	+	+	+		+	+
26	ZmABCG26	ABCG28	Zm00001d046068	9	1040						+				
27	ZmABCG27	ms2	Zm00001d046537	9	676	+		+	+	+		+	+	+	+
28	ZmABCG28	-	Zm00001d024075	10	892					+		+			+
29	ZmABCG29	-	Zm00001d024497	10	727	+	+	+	+	+		+		+	
30	ZmABCG30	ABCG24	Zm00001d025031	10	1101	+	+	+	+	+		+			
31	ZmABCG31	-	Zm00001d026706	10	750	+	+	+	+	+	+		+	+	+
32	ZmABCG32	-	Zm00001d031724	1	1357			+		+				+	
33	ZmABCG33	-	Zm00001d034940	1	1490	+	+	+	+	+			+	+	+
34	ZmABCG34	ABCG51	Zm00001d003354	2	1425			+	+					+	+
35	ZmABCG35	glossy13	Zm00001d039631	3	1425		+	+		+	+	+		+	
36	ZmABCG36	ABCG38	Zm00001d043598	3	1488	+		+							+
37	ZmABCG37	-	Zm00001d044442	3	1574	+		+	+	+			+	+	
38	ZmABCG38	ABCG37	Zm00001d044443	3	1448	+		+					+		
39	ZmABCG39	ABCG39	Zm00001d048860	4	1446	+		+	+				+	+	+
40	ZmABCG40	ABCG45	Zm00001d052726	4	1237	+									
41	ZmABCG41	ABCG39	Zm00001d053612	4	1443	+		+	+	+					
42	ZmABCG42	ABCG42	Zm00001d036986	6	1508	+		+		+					+
43	ZmABCG43	ABCG45	Zm00001d019951	7	1224	+									
44	ZmABCG44	-	Zm00001d019961	7	1056	+		+	+						
45	ZmABCG45	ABCG53	Zm00001d020134	7	1451	+		+		+					
46	ZmABCG46	ABCG43	Zm00001d021647	7	1445	+		+	+						
47	ZmABCG47	ABCG36	Zm00001d011315	8	1439	+		+	+		+	+		+	
48	ZmABCG48	ABCG37	Zm00001d011319	8	1449	+									
49	ZmABCG49	ABCG42	Zm00001d046277	9	1540	+		+	+	+					
50	ZmABCG50	ABCG41	Zm00001d023392	10	1472	+									
51	ZmABCG51	ABCG48	Zm00001d025012	10	1465	+		+	+		+				

Notes: ^1^. Synonyms were retrieved from MaizeGDB (www.maizegdb.org (accessed on 8 May 2022)) and NCBI (https://www.ncbi.nlm.nih.gov/ (accessed on 8 May 2022)); ^2^. Expression information of maize ABCG genes was retrieved from MaizeGDB (www.maizegdb.org (accessed on 20 April 2022)).

**Table 2 ijms-23-09304-t002:** Functional Classifications of The Reported ABCG Genes in *Arabidopsis*, Rice and Their Orthologs in Maize.

No.	Gene Name	Gene ID	Maize Ortholog	Expression Organ	Biological Function	Reference
**I. Anther development and male fertility**						
1	*AtABCG11*	AT1G17840	*ZmABCG13* *ZmABCG3* *ZmABCG6*	Stems; Leaves; Roots; Inflorescences; Flowers; Seeds; Siliques	Flower cutin and root suberin formation; Vascular development.	[17,25,26]
2	*OsABCG26*	Os10g0494300	*ZmABCG13* *ZmABCG3*	Anthers; Pistils	Anther cuticle and pollen exine formation and pollen-pistil interaction	[18,19]
3	*ZmMs13/* *ZmABCG13 **	Zm00001d013960	-	Anthers	Callose dissolution; Anther cuticle and pollen exine formation	[7]
4	*AtABCG26*	AT3G13220	*ZmABCG27*	Anthers; Siliques; Leaves; Stems	Exine formation	[12,13,14]
5	*OsABCG15*	Os06g0607700	*ZmABCG27*	Anthers	Exine formation and pollen development	[15,16,19]
6	*ZmMs2/* *ZmABCG27 ***	Zm00001d046537	-	Anthers	Transport of anther cutin and wax components	[8,10]
7	*AtABCG1*	At2G39350	*ZmABCG2*	Roots; Flowers; Stamens; Pistils; Seedlings; Leaves; Shoot meristems	Male gametophyte development; Pollen tube growth; Suberin formation in roots.	[20,21,22,27]
8	*AtABCG16*	AT3G55090	*ZmABCG2*	Cotyledons; Roots; Seeds; Flowers; Leaves	Male gametophyte development and pollen tube growth; Anther filament; JA and ABA responses; Plant pathogen response.	[20,21,22,28,29]
9	*OsABCG3/* *LSP1*	Os01g0836600	*ZmABCG24* *ZmABCG8*	Anthers	Normal pollen fertility and pollen wall formation	[23,24]
10	*AtABCG9*	AT4G27420	*ZmABCG29*	Cotyledons; Leaves; Stamen; Siliques; Roots; Anthers	Sterol accumulated on pollen surface; Proper vascular development	[11,25]
11	*AtABCG31*	AT2G29940	*ZmABCG34*	Inflorescences; Anthers; Rosette leaves; Stems; Seedlings; Siliques	Sterol accumulation on pollen surface; ABA export from endosperm; Seed germination; Disease response	[11,30]
12	*AtABCG28*	AT5G60740	*ZmABCG26* *ZmABCG14*	Pollen tubes; Pollen grains	Polyamine and ROS translocation at the growing tip of pollen tube	[31]
**II. Vegetative and female organ development**						
1	*AtABCG2*	AT2G37360	*ZmABCG2*	Roots; Seedlings; Anthers	Suberin barrier synthesis in roots and seed coats	[21]
2	*AtABCG6*	AT5G13580	*ZmABCG2*	Roots; Seeds; Anthers	Suberin barrier synthesis in roots and seed coats	[21]
3	*AtABCG20/* *AtAwake1*	AT3G53510	*ZmABCG2*	Roots; Seedings and anthers	Suberin barrier synthesis in roots and seed coats; Seed dormancy	[21,32]
4	*AtABCG5*	AT2G13610	*ZmABCG31*	Cotyledons; Roots; Shoots	Wax precursor transport; Shoot branching; Root suberization	[33,34,35]
5	*AtABCG12*	AT1G51500	*ZmABCG21*	Stems; Leaves; Siliques; Flowers; Roots	Cuticle formation; Wax secretion; Abiotic stress response	[26,36]
6	*AtABCG13*	AT1G51460	*ZmABCG21*	Flowers; Leaves; Stems; Siliques; Roots	Flower cuticular lipids transport	[37,38]
7	*AtABCG21*	AT3G25620	*ZmABCG29*	Seedlings; Leaves	Stomatal regulation	[39]
8	*AtABCG27*	AT3G52310	*ZmABCG1*	Flowers; Pistils; Leaves	Cellulose synthesis; Flower and leaf development	[40,41]
9	*AtABCG29*	AT3G16340	*ZmABCG36*	Roots; Stems; Leaves; Anthers; Siliques; Seedings	Lignin biosynthesis	[40,42]
10	*AtABCG32*	AT2G26910	*ZmABCG35*	Leaves; Stems; Flowers; Seedlings; Siliques	Cuticular layer of the cell wall formation	[43,44]
11	*OsABCG31*	Os01g0177900	*ZmABCG35*	Leaves	Cuticle formation	[45]
12	*AtABCG33*	AT2G37280	*ZmABCG50*	Stems; Roots	Cellulose synthesis; Lignification	[40,46]
13	*AtABCG34*	AT2G36380	*ZmABCG46* *ZmABCG33* *ZmABCG51*	Roots; Leaves	Camalexin secretion to leaf surface and thereby prevents *A. brassicicola* infection; Root exudation	[47]
14	*AtABCG37*	AT3G53480	*ZmABCG50*	Roots; Seedlings	IBA transported out of the cells; Secretion of scopoletin and derivatives	[48,49]
15	*ZmGL13*	Zm00001d039631	*-*	Leaves	Necrotic glossy leaf; Plants smaller than nonmutant sibs	[9]
**III. Biotic and abiotic stress response**						
1	*OsABCG5/* *RCN1*	Os03g0281900	*ZmABCG2*	Roots; Tiller buds; Basal part of stem; Leaves.	Lateral shoot outgrowth; ABA accumulation in guard cells under drought stress; Root hypodermis suberization	[34,35,50]
2	*AtABCG19*	AT3G55130	*ZmABCG2*	Leaves; Roots; Flowers; Seedings	Kanamycin resistance; Zinc homeostasis; Nicotianamine transport	[51,52]
3	*OsABCG9*	Os04g0528300	*ZmABCG6*	Seminal roots; Young shoots; Anthers; Stems	Cuticular permeability and drought sensitivity	[53]
4	*OsABCG43*	Os07g0522500	*ZmABCG46*	Roots	Cd tolerance in yeast	[54]
5	*AtABCG35*	AT1G15210	*ZmABCG36*	Roots	Root exudation; Cadmium response	[55,56]
6	*AtABCG36*	AT1G59870	*ZmABCG36*	Roots; Leaves; hydathodes; Stems; Inflorescences; Flowers; Siliques	An efflux pump of Cd^2+^ or Cd conjugates; Drought and salt resistance; IBA Transport; Root and cotyledon development	[57,58,59]
7	*OsABCG36*	Os01g0609300	*ZmABCG36*	Roots; Shoots	Cadmium tolerance; Heavy metal stress	[60]
8	*AtABCG40*	AT1G15520	*ZmABCG41/* *ZmABCG39*	Inflorescences; Flowers; Roots; Leaves; Stems; Seeds; Siliques	Stomatal regulation and ABA importation; Seed germination; Pathogen and drought response	[30,61,62,63]
9	*OsSTR1*	Os09g0401100	*ZmABCG28*	Roots; Panicles; Leaves; Stems; Embryos	Mycorrhizal arbuscule formation	[64]
10	*OsSTR2*	Os07g0191600	*-*			
**IV. Hormone transport and signaling**						
1	*AtABCG14*	AT1G31770	*ZmABCG29*	Cotyledons; Rosette leaves; Flowers; Roots; Siliques	Vascular development; Cytokinin translocation in shoot	[25,65]
2	*OsABCG18*	Os08g0167000	*ZmABCG29*	Roots; Stems; Leaves; Panicles	Long-distance transport of cytokinin in shoot and promote grain yield	[66]
3	*AtABCG17*	AT3G55100	*ZmABCG2*	Roots; Leaves	ABA homeostasis and long-distance translocation	[67]
4	*AtABCG18*	AT3G55110				
5	*AtABCG22*	AT5G06530	*ZmABCG1*	Seedlings; Roots; Stems; Leaves; Flowers	Stomatal regulation; ABA signaling; Lignification	[39,40,68]
6	*AtABCG25*	AT1G71960	*ZmABCG12/* *ZmABCG7*	Seedlings; Roots; Stems; Leaves; Flowers	Stomatal regulation; ABA export from endosperm; Seed germination	[30,62,69]
7	*AtABCG30*	AT4G15230	*ZmABCG50*	Roots; Seeds	ABA import into the embryo; Seed germination; Monolignol transport	[30,55]

Notes: At, *Arabidopsis*; Os, rice; Zm, Maize. * Previously named as ZmABCG2a; ** previously named as ZmABCG26. II. 16–18, AtABCG1, AtABCG9, and AtABCG11 are also involved in vegetative and female organ development. IV. 8-13, AtABCG31, AtABCG36, AtABCG37, AtABCG40 and OsABCG5/RCN1 are also involved in hormone response.

**Table 3 ijms-23-09304-t003:** The Substrate Identification of Plant ABCG Transporters.

No.	Transporters	Types	Substrate(s)	Method of Substrate Identification	Reference
**I. Isotope labelling experiment in vivo**					
1	AtABCG11	Half-size	10,16-diOH C16:0-2-glycerol; ω-OH C16:0	[^3^H]-10,16-diOH C16:0-2-glycerol; [^14^C]-u-OH C16:0 export assay using *N. benthamiana* protoplast system	[44]
2	AtABCG14	Half-size	Cytokinins	WT and *atabcg14* mutant cultivated in ^14^C-labeled trans-zeatin (tZ) medium	[65]
3	AtABCG16	Half-size	Jasmonate	^3^H-JA transport of yeast strain expressing AtABCG16; transport of ^3^H-JA and ^3^H-JA-Ile by the nuclei isolated from *abcg16* plant	[28]
4	AtABCG25	Half-size	ABA	^3^H-ABA transport of yeast strains expressing AtABCG25	[30,69]
5	AtABCG30	Full-size	ABA	^3^H-ABA transport of yeast strains expressing AtABCG30	[30]
6	AtABCG31	Full-size	ABA	^3^H-ABA transport of yeast strains expressing AtABCG31	[30]
7	AtABCG32	Full-size	10,16-diOH C16:0-2-glycerol; ω-OH C16:0; C16:0 DCA	[^3^H]-10,16-diOH C16:0-2-glycerol; [^14^C]-u-OH C16:0; [^14^C]-C16:0 DCA export assay using *N. benthamiana* protoplast system	[44]
8	AtABCG37	Full-size	Auxin precursor indole-3-butyric acid (IBA)	^3^H-IBA export from *abcg37* leaf mesophyll protoplasts; ^3^H-IBA transport in yeast strains expressing AtABCG37; export of ^3^H-2,4-D and ^3^H-IBA in HeLa cells expressing AtABCG37	[49]
9	AtABCG40	Full-size	ABA	^3^H-ABA uptake in yeast strain YMM12 and BY-2 cell lines expressing AtABCG40	[30]
**II. Substance analysis using the GC-MS system**					
1	AtABCG2	Half-size	Suberin precursors	Suberin monomers analysis by using GC-MS system	[21]
2	OsABCG5/ RCN1	Half-size	ABA; suberin monomers	Phenotypic and expression analysis of RCN1 in WT and RCN1-RNAi plants treated with ABA; Histochemical staining and suberin contents analysis by GC-MS	[34,35,50]
3	AtABCG5	Half-size	Cutin/wax precursors	Substance analysis by using GC-MS	[33]
4	AtABCG6	Half-size	Fatty acids; Fatty alcohols	Suberin monomers analysis by using GC-MS	[21]
5	AtABCG9	Half-size	Steryl glycosides	Substance analysis by using GC-MS	[11]
6	OsABCG9	Half-size	Wax precursors	Substance analysis by using GC-MS	[53]
7	OsABCG26	Half-size	Anther cuticular wax and cutin monomers	Substance analysis by using GC-MS	[18,19]
8	ZmMS13/ ZmABCG13	Half-size	Anther cuticular wax and cutin monomers	Substance analysis by using GC-MS	[7]
9	AtABCG12	Half-size	Lipids; Wax components	Substance analysis by using GC-MS	[26,36]
10	AtABCG13	Half-size	Cuticular lipids	Substance analysis by using GC-MS	[37]
11	AtABCG20/ Awake1	Half-size	Suberin precursors; Fatty acids	Substance analysis by using GC-MS	[21,32]
12	AtABCG26	Half-size	Sporopollenin precursors; Polyketides	Substance analysis by using GC-MS	[12,14]
13	OsABCG15	Half-size	Sporopollenin and cutin precursors	Substance analysis by using GC-MS	[15,16]
14	ZmMS2/ ABCG26	Half-size	Sporopollenin and cutin precursors	Substance analysis by using GC-MS	[8,10]
15	OsABCG31	Full-size	Cutin precursors	Substance analysis by using GC-MS	[45]
**III. Other methods**					
1	AtABCG1	Half-size	Fatty alcohols; Fatty acids	AtABCG1 protein purified in *Pichia pastoris* to test its ATPase assay	[27]
2	ZmMS13/ ZmABCG13	Half-size	Fatty alcohols; Fatty acids	ZmMS13 protein purified in *E. coli* and its ATPase assay in vitro	[7]
3	OsABCG18	Half-size	Cytokinins	Export assay with OsABCG18 expressing *N. benthamiana* protoplasts	[66]
4	OsABCG3/ LSP1	Half-size	Pollen wall and coat materials	Phenotypic and qRT-PCR analysis of WT and *osabcg3* mutant	[23,24]
5	AtABCG19	Half-size	Kanamycin	Kanamycin treatment of AtABCG19-overexpressing plants	[52]
6	AtABCG22	Half-size	ABA; lignin precursors	Speculation according to the same phenotype with AtABCG21	[40,68]
7	AtABCG28	Half-size	Polyamine	Immunostaining of polyamines in pollen tubes	[31]
8	AtABCG29	Full-size	p-coumaryl alcohol	p-coumaryl alcohol uptake using microsomes from S. *cerevisiae* expressing AtABCG29	[42]
9	AtABCG34	Full-size	Camalexin	Camalexin toxicity assay in *Arabidopsis* and BY2 cells expressing AtABCG34	[47]
10	OsABCG43	Full-size	Cd	Cd inducible and confers Cd tolerance on OsABCG43-expressing yeast cell.	[54]
11	AtABCG36	Full-size	Lignin precursors; Cd; IBA	The *atabcg36* mutants and WT were tested in IBA-containing agar medium under yellow filtered light conditions.	[57,59]
12	OsABCG36	Full-size	Cd^2+^	Efflux transport of Cd in OsABCG36-expressing yeast cell.	[60]

## Data Availability

All data are shown in the main manuscript and in the Appendix A.

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
