# Peer review of "ATP-Binding Cassette G Transporters and Their Multiple Roles Especially for Male Fertility in Arabidopsis, Rice and Maize"

_ijms, 2022, doi:10.3390/ijms23169304_

Round 1

Reviewer 1 Report

This review describes ABCG plant transporter genes in Arabidopsis, Rice and Maize. It covers most of new and previous works, describes genes phylogenetically, functionally and by expression in different organs. This is really a comprehensive review and I have nothing to suggest. May be only reformat in Table 1 the expression column:  to make organ names as columns titles and expression for each gene as “+”, or a colored box in the respective intersection. This way the organ-related expression pattern will be clearly visible. Generally, the paper can be published as is.

Author Response

Reviewer #1:
This review describes ABCG plant transporter genes in Arabidopsis, Rice and Maize. It covers most of new and previous works, describes genes phylogenetically, functionally and by expression in different organs. This is really a comprehensive review and I have nothing to suggest. May be only reformat in Table 1 the expression column:  to make organ names as columns titles and expression for each gene as “+”, or a colored box in the respective intersection. This way the organ-related expression pattern will be clearly visible. Generally, the paper can be published as is.

Response: Thank you for the positive comments on our manuscript. We have revised Table 1 according to your suggestion and made organ names as columns titles and expression for each gene as “+” (in Pages 3-4).

Reviewer 2 Report

Minor remarks are marked in attached file.

Author Response

Reviewer #2:

Minor remarks are marked in attached file.

Response: Thank you for the kind comments.

  1. Follow your suggestions, we have revised the manuscript carefully as below.

The “serval” has been changed to “several” in line 232.

The “Zinc” has been changed to “zinc” in line 241.

The “17” has been changed to “14” in line 250.

The “GMS” has been changed to “genic male sterile (GMS)” in line 290.

  1. What is the source of difference in maize ABCG number mentioned in Introduction and in this section? An alternative splicing?

Response: The different number of ABCG genes may be resulted from an alternative splicing of three ABCGs. There are 54 ABCG gene models predicted in maize genome (B73 V3) in a previous study (Pang et al., 2013). However, by searching in maize genome (B73 V4), we find that 51 ABCG gene models in maize genome, among which ZmABCG5, ZmABCG7 and ZmABCG12 correspondence to two gene models in maize genome B73 V3, respectively (Table S1). Therefore, we update the maize ABCG gene number as 51 in this review. Furthermore, we revised the manuscript as follow: “Notably, our identified ABCG gene number (51) is different from that (54) in the previous report [6], which may be resulted from an alternative splicing of three ABCG genes including ZmABCG5, ZmABCG7 and ZmABCG12 (Table S1)” in lines 77-79.

  1. Is this true for all cluster II genes or only for six of them shown in Figure 4 B10-B15?

Response: This is true for all the eight genes in cluster II. Considering that the expression patterns of ZmABCG13/ZmMs13 and ZmABCG27/ZmMs2 have been reported in previous study (Fang et al., 2022; Jiang et al., 2021; Xu et al., 2021), we only showed the qRT-PCR expression patterns of the other six ABCG genes of cluster II in Figure 4 B10-B15. In the revised manuscript, we cited the references before Figure 4 B10-B15 (in line 318).